# A Simple Approach to Case-Based Reasoning in Knowledge Bases

**Rajarshi Das**[1]                                                        RAJARSHI@CS.UMASS.EDU
**Ameya Godbole**[1]                                                        AGODBOLE@CS.UMASS.EDU
**Shehzaad Dhuliawala**[2]                                        SHEHZAAD.DHULIAWALA@MICROSOFT.COM
**Manzil Zaheer**[3]                                                     MANZILZAHEER@GOOGLE.COM
**Andrew McCallum**[1]                                                     MCCALLUM@CS.UMASS.EDU

[1]*University of Massachusetts, Amherst*
[2]*Microsoft Research, Montreal*
[3]*Google Research*

## Abstract

We present a surprisingly simple yet accurate approach to reasoning in knowledge graphs (KGs) that requires *no training*, and is reminiscent of case-based reasoning in classical artificial intelligence (AI). Consider the task of finding a target entity given a source entity and a binary relation. Our non-parametric approach derives crisp logical rules for each query by finding multiple *graph path patterns* that connect similar source entities through the given relation. Using our method, we obtain new state-of-the-art accuracy, outperforming all previous models, on NELL-995 and FB-122. We also demonstrate that our model is robust in low data settings, outperforming recently proposed meta-learning approaches[1].

## 1. Introduction

Given a new problem, humans possess the innate ability to 'retrieve' and 'adapt' solutions to *similar* problems from the past. For example, an automobile mechanic might fix a car engine by recalling previous experiences where cars exhibited similar symptoms of damage. This model of reasoning has been widely studied and verified in cognitive psychology for various applications such as mathematical problem solving [Ross, 1984], diagnosis by physicians [Schmidt et al., 1990], automobile mechanics [Lancaster and Kolodner, 1987] etc. A lot of classical work in artificial intelligence (AI), particularly in the field of *case-based reasoning* (CBR) has focused on incorporating such kind of reasoning in AI systems [Schank, 1982, Kolodner, 1983, Rissland, 1983, Aamodt and Plaza, 1994, Leake, 1996, inter-alia].

At a high level, a case-based reasoning system is comprised of four steps [Aamodt and Plaza, 1994] — (a) 'retrieve', in which given a new problem, 'cases' that are similar to the given problem are retrieved. A 'case' is usually associated with a problem description (used for matching it to a new problem) and its corresponding solution. After the initial retrieval step, the previous solutions are (b) 'reused' for the problem in hand. Often times, however, the retrieved solutions cannot be directly used, and hence the solutions needs to be (c) 'revised'. Lastly, if the revised solution is useful for solving the given problem, they are (d) 'retained' in a memory so that they can be used in the future.

Knowledge graphs (KGs) [Suchanek et al., 2007, Bollacker et al., 2008, Carlson et al., 2010] contain rich facts about entities and capture relations with diverse semantics between those entities. However, KGs can be highly incomplete [Min et al., 2013], missing important edges (relations)

---

[1]Code available at https://github.com/rajarshd/CBR-AKBC

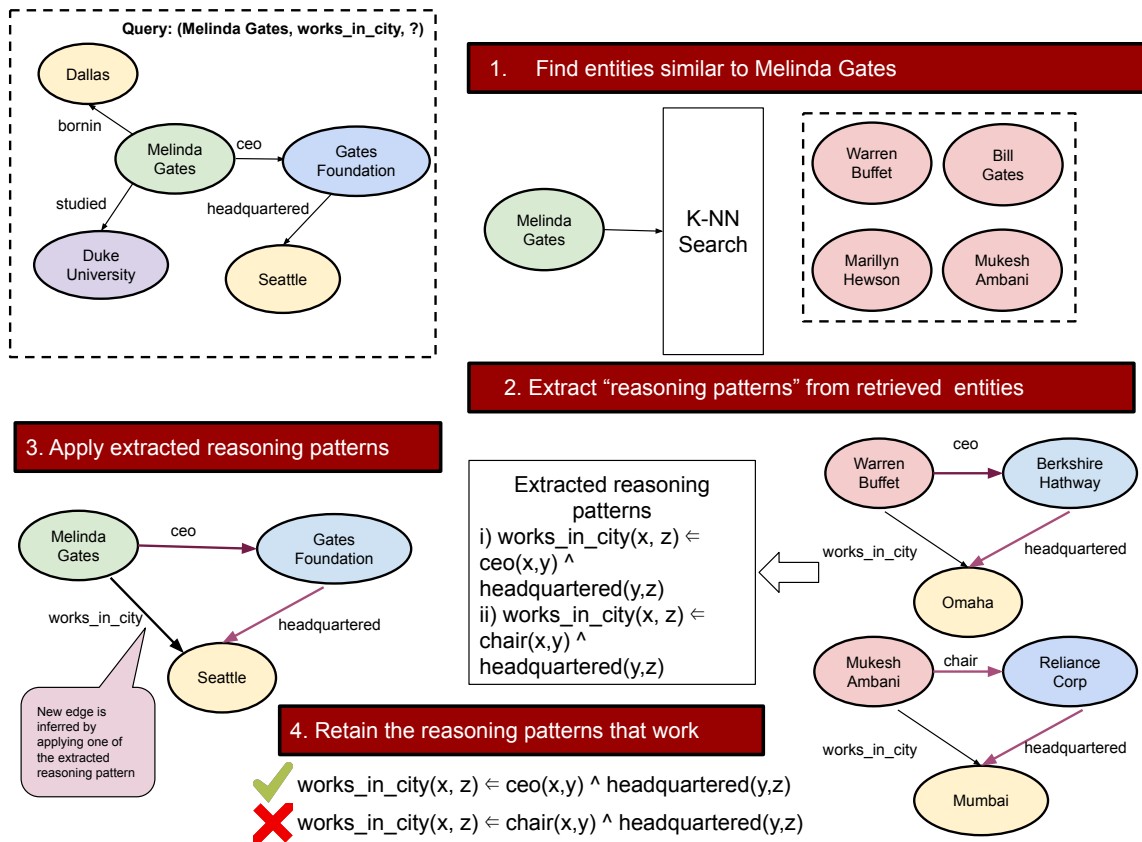

Figure 1: Overview of our approach. Given a query (Melinda, works_in_city, ?), our method first retrieves similar entities to the query entity. Then it gathers the reasoning paths that lead to the answer for the respective retrieved entities. The reasoning paths are applied for the query entity (Melinda) to retrieve the answer.

between entities. For example, consider the small fragment of the KG around the entity MELINDA GATES in figure 1. Even though a lot of facts about MELINDA is captured, it is missing the edge corresponding to works_in_city. A recent series of work address this problem by modeling multi-hop paths between entities in the KG, thereby able to reason along the path MELINDA → ceo → GATES FOUNDATION → headquartered → SEATTLE. However, the number of paths starting from an entity increases exponentially w.r.t the path length and therefore past work used parametric models to do approximate search using reinforcement learning (RL) [Xiong et al., 2017, Das et al., 2018, Lin et al., 2018]. Using RL-based methods have their own shortcomings, like hard to train and high computational requirements. Moreover, these models try to encode all the rules for reasoning into the parameters of the model which makes learning even harder.

In this paper, we propose a simple non-parametric approach for reasoning in KGs (Figure 1). Given an entity and a query relation $(e_q, r_q)$, we first retrieve $k$ entities in the KG that are similar to $e_q$ and for which we observe the query relation edge $r_q$. The retrieved entities could be present anywhere in the KG and are not just restricted in the immediate neighborhood of $e_q$. Similarity between entities is measured based on the observed relations that the entities participate in. This ensures that the retrieved entities have similar observed properties as $e_q$ (e.g., if $e_q$ is a CEO, then the retrieved entities are also CEO's or business person). Next, for each of the retrieved entities, our

method finds a set of reasoning paths that connect the retrieved entities to the entities that they are connected with via the query relation $r_q$. In this way, our method removes the burden of storing the reasoning rules in the parameters of the model and rather extract it from entities similar to the query entity. Next, our method checks if similar reasoning paths exists starting from the query entity $e_q$. If similar paths exists, then the answer to the original query is found by starting from $e_q$ and traversing the KG by following the reasoning path. In practice, we find *multiple* reasoning paths which end at different entities and we rank the entities based on the number of reasoning paths that lead to them, with the intuition that an entity supported by multiple reasoning paths is likely to be a better answer to the given query.

Apart from being non-parametric, our proposed method has many other desirable properties. Our method is generic and even though we find very simple and symbolic methods to work very well for many KG datasets, every component of our model can be augmented by plugging in sophisticated neural models. For example, currently we use simple symbolic string matching to find if a relation path exists for the query entity. However, this component can be replaced with a neural model that matches paths which are semantically similar to each other [Das et al., 2017]. Similarly, we currently use a very simple inner product to compute similarities between entity embeddings. But that can be replaced with more sophisticated maximum inner product search [Mussmann and Ermon, 2016].

The contributions of the paper are as follows — (a) We present a non-parametric approach for reasoning over KGs, that uses *multiple* paths of evidence to derive an answer. These paths are gathered from entities in different parts of the KG that are similar to the query entity. (b) Our approach requires *no training* and can be readily applied to any new KGs. (c) Our method achieves state-of-the-art performance on NELL-995 and the harder subset of FB-122 outperforming sophisticated neural approaches and other tensor factorization methods. We also perform competitively on the WN18RR dataset. (d) Lastly, we provide detailed analysis about why our method outperforms parametric rule learning approaches like MINERVA [Das et al., 2018].

## 2. Method

### 2.1 Notation and Task Description

Let $\mathcal{E}$ denote the set of entities and $\mathcal{R}$ denote the set of binary relations. A knowledge base (KB) is a collection of facts stored as triplets $(e_1, r, e_2)$ where $e_1, e_2 \in \mathcal{E}$ and $r \in \mathcal{R}$. From the KB, a knowledge graph $\mathcal{G}$ can be constructed where the entities $e_1, e_2$ are represented as the nodes and relation $r$ as labeled edge between them. Formally, a KG is a directed labeled multigraph $\mathcal{G} = (V, E, \mathcal{R})$, where $V$ and $E$ denote the vertices and edges of the graph respectively. Note that $V = \mathcal{E}$ and $E \subseteq V \times \mathcal{R} \times V$. Also, following previous approaches [Bordes et al., 2013, Xiong et al., 2017], we add the inverse relation of every edge, i.e. for an edge $(e_1, r, e_2) \in E$, we add the edge $(e_2, r^{-1}, e_1)$ to the graph. (If the set of binary relations $\mathcal{R}$ does not contain the inverse relation $r^{-1}$, it is added to $\mathcal{R}$ as well). A path in a KG between two entities $e_s$, $e_t$ is defined as a sequence of alternating entity and relations that connect $e_s$ and $e_t$. A length of a path is the number of relation (edges) in the path. Formally, let a path $p = (e_1, r_1, e_2, \ldots, r_n, e_{n+1})$ with $st(p) = e_1$, $en(p) = e_{n+1}$ and $len(p) = n$. Let $\mathcal{P}$ denote the set of all paths and $P_n$ denote the set of all paths with length up to $n$, i.e. $P_n \subseteq \mathcal{P} = \{p \in \mathcal{P} \mid len(p) \leq n\}$.

We consider the task of query answering on KGs. Query answering seeks to answer questions of the form $(e_{1q}, r_q, ?)$ (e.g. Melinda, works_in_city, ?), where the answer is an entity in the KG.

## 2.2 Case-based Reasoning on Knowledge Graphs

This section describes how we apply case-based reasoning (CBR) in a KG. In CBR, given a new problem, similar cases are retrieved from memory [Aamodt and Plaza, 1994]. In our setting, a problem is a query $(e_{1q}, r_q, ?)$ for a missing edge in the KG. A case is defined as an observed fact $(e_1, r, e_2)$ in a KG along with a set of paths up to length $n$ that connect $e_1$ and $e_2$. Formally, a case $c = (e_1, r, e_2, P) \subseteq V \times \mathcal{R} \times V \times P_n$ is a 4 tuple where $(e_1, r, e_2) \in \mathcal{G}$ and $P \subseteq P_{(e_1, e_2)} = \{p \in P_n \mid st(p) = e_1, en(p) = e_2\}$. In practice, it is intractable to store *all* paths between $e_1$ and $e_2$ ($P_{(e_1, e_2)}$) and therefore we only store a random sample. Note, that we have a case for every observed triple in the KG and we denote the set of all such cases as $\mathcal{C}$. We also assume access to a pre-computed similarity matrix $S \in \mathbb{R}^{V \times V}$ which stores the similarity between any two given entities in a KG. Intuitively, two entities which exhibit the same relations should have a high similarity (e.g., an athlete should have a high similarity score with another athlete). Lastly, we define a memory $\mathcal{M}$ that serves as the store for all the cases present in a KG and also the similarity matrix, i.e. $\mathcal{M} = (\mathcal{C}, S)$.

As explained earlier, CBR broadly comprises of four steps, which we explain briefly for our setup on a KG.

- **Retrieve**: In this step, given a query $(e_{1q}, r_q, ?)$, our method first retrieves a set of $k$ similar entities w.r.t $e_{1q}$ using the pre-computed similarity matrix $S$, such that for each entity $e'$ in the retrieved set, we observe at least one fact of the form $(e', r_q, e'')$ in $\mathcal{G}$. In other words, each retrieved entity should have at least one outgoing edge in $\mathcal{G}$ with the edge label as $r_q$. Next, we gather all such facts $(e', r_q, e'')$ in $\mathcal{G}$ for each of the retrieved $e'$. Note, that there could be more than one fact for a given entity and query relation (e.g. USA, has_city, New York City and USA, has_city, Boston, for the entity 'USA' and query relation 'has_city'). Finally for all the gathered facts from the $k$-nearest neighbors, we retrieve all the cases from the memory store $\mathcal{M}$. As noted before, in our formulation, a case is a fact augmented with a sample of KG paths that connect the entities of the fact. As we will see later (§ 3), the KG paths often represent reasoning rules that determine why the fact $(e', r_q, e'')$ hold true. The goal of CBR is to reuse these rules for the new query.

- **Reuse**: The reasoning paths that were gathered in the retrieve step are re-used for the query entity. As described before, a path in a KG is an alternating sequence of entities and relations. The path gathered from the nearest neighbors have entities which are in the immediate neighborhood of the nearest neighbor entities. To reuse these paths, we first replace the entities from the paths with un-instantiated variables and only extract the sequence of relations in them [Schoenmackers et al., 2010]. For example, if we have a retrieved case (a fact with a set of paths) such as $((\text{USA}, \text{has\_city}, \text{Boston}), \{(\text{USA}, \text{has\_state}, \text{Massachusetts}, \text{city\_in\_state}, \text{Boston})\})$, we remove the entities from the path and extract rules such as: $\text{has\_state}(x, y), \text{city\_in\_state}(y, z) \implies \text{has\_city}(x, z)$. We gather all such paths from all the cases retrieved for a query. Since the same path type can occur in different cases, we maintain a list of paths sorted w.r.t the counts (in descending order).

- **Revise**: After the paths have been gathered, we look if those paths exists for the query entity $e_{1q}$. In our approach we find that simple symbolic exact-string matching for the relations works quite well. However, neural relation extraction systems [Zeng et al., 2014, Verga et al., 2016] can be incorporated to map a relation to other similar relations to improve recall. We keep this direction as a part of our future work. Instead if we find an exact match for the sequence of

relations, we revise the rules by instantiating the variables with the entities which lie along the path in the neighborhood of $e_{1q}$.

- **Retain**: Finally, a case including the query fact and the paths that lead to the correct answer for the query can be added to the memory store $\mathcal{M}$.

**Computing the similarity matrix**: CBR approaches need access to a similarity matrix S to retrieve similar cases to the query entity. Intuitively, the similarity between entities that have similar relations should be higher (e.g. similarity between two athletes should be higher than the similarity between an athlete and a country). To model this, we parameterize each entity with an *m*-hot vector $\mathbf{e} \in \mathbb{R}^{\mathcal{R}}$. That is, each entity is a *m*-hot vector with the dimension equal to the size of the number of binary relations in the KB. An entry in the vector is set to 1, if an entity has at least one edge with that relation type, otherwise is set to 0. Even though, this is a really simple way of parameterizing an entity, we found this to work extremely well in practice. However, as previously mentioned we propose a generic method and one could replace the entity embeddings with any pre-trained vectors from any model. As an example, we present experiments by replacing our *m*-hot representation with pre-trained embeddings obtained from the state-of-the-art RotatE model [Sun et al., 2019]. Lastly, the similarity between two entities is calculated by a simple inner product between the normalized embeddings.

**Caching 'cases' in the memory store**: CBR also needs access to store containing cases. As mentioned before, in our setup a 'case' is a KG triple, along with a sample of paths that connect the two entities of the triple. Since the number of paths between entities grow exponentially w.r.t path length, it is intractable to store all paths between an entity pair. We instead consider a small subgraph around each entity in the KG spanned by 1000 randomly sampled paths of length up to 3. Next, for each triple $(e_1, r, e_2)$ in the KG, we exhaustively search the subgraph around $e_1$, collected in the previous step to find paths up to length 3 which lead to $e_2$. These paths along with the fact form a case. This process is repeated for all the triples and each case is added to the case store $\mathcal{C}$.

Both the similarity matrix $S$ and the case store $\mathcal{C}$ are pre-computed offline and is stored in the memory $\mathcal{M}$. Once that is done, our method requires *no further training* and can be readily used for any query in the KG.

## 3. Experiments

### 3.1 Data and Evaluation Protocols

We test our CBR based approach on three datasets that are often used in the community for benchmarking models — FB122 [Guo et al., 2016], NELL-995 [Xiong et al., 2017], and WN18RR [Dettmers et al., 2018]. FB122 comes with a set of KB rules that can be used to infer missing triples in the dataset. We do not use the rules in the dataset and even show that our model is able to recover the rules from similar entities to the query. WN18RR was created by Dettmers et al. [2018] from the original WN18 dataset by removing various sources of test leakage, making the datasets more realistic and challenging. We compare our CBR based approach with various state-of-the-art models using standard ranking metrics such as HITS@N and mean reciprocal rank (MRR). For fair comparison to baselines, after a fact is predicted we do not add the new case (inferred fact and paths) in the memory (retain step in § 2.2), as that would mean we would use more information than our baselines to predict the followup queries.

Hyper-parameters: The various hyper-parameters for our method are the number of nearest neighbor retrieved for a query entity ($k$), the number of paths that are gathered from the retrieved entity ($l$) and

| | Model | HITS@3 | HITS@5 | HITS@10 | MRR |
|---|---|---|---|---|---|
| **WITH RULES** | KALE-Pre [Guo et al., 2016] | 0.358 | 0.419 | 0.498 | 0.291 |
| | KALE-Joint [Guo et al., 2016] | 0.384 | **0.447** | **0.522** | 0.325 |
| | *ASR*-DistMult [Minervini et al., 2017] | 0.363 | 0.403 | 0.449 | 0.330 |
| | *ASR*-ComplEx [Minervini et al., 2017] | 0.373 | 0.410 | 0.459 | 0.338 |
| **WITHOUT RULES** | TransE [Bordes et al., 2013] | 0.360 | 0.415 | 0.481 | 0.296 |
| | DistMult [Yang et al., 2017] | 0.360 | 0.403 | 0.453 | 0.313 |
| | ComplEx [Trouillon et al., 2016] | 0.370 | 0.413 | 0.462 | 0.329 |
| | GNTPs [Minervini et al., 2020] | 0.337 | 0.369 | 0.412 | 0.313 |
| | CBR (Ours) | **0.424** | **0.471** | **0.515** | **0.378** |

Table 1: Link prediction results on the FB122 dataset.

| Metric | ComplEx | ConvE | DistMult | MINERVA | CBR |
|---|---|---|---|---|---|
| HITS@1 | 0.612 | 0.672 | 0.610 | 0.663 | **0.705** |
| HITS@3 | 0.761 | 0.808 | 0.733 | 0.773 | **0.828** |
| HITS@10 | 0.827 | 0.864 | 0.795 | 0.831 | **0.875** |
| MRR | 0.694 | 0.747 | 0.680 | 0.725 | **0.772** |

Table 2: Query-answering results on NELL-995 dataset.

the maximum path length considered ($n$). For all our experiments, we set $n = 3$. We tune $k$ and $l$ for each dataset w.r.t the given validation set.

### 3.2 Results on Query Answering

We first present results on query answering (link prediction) on the three datasets and compare to various state-of-the-art baseline models. It is quite common in literature [Bordes et al., 2013, Yang et al., 2015, Dettmers et al., 2018, Sun et al., 2019] to report aggregate results on both tail prediction ($e_1, r, ?$) and head prediction ($?, r^{-1}, e_2$). To be exactly comparable to baselines, we report results on tail prediction for NELL-995 and for other datasets, we report average of head and tail predictions.

**NELL-995**: Table 2 reports the query answering performance on the NELL-995 dataset. We compare to several strong baselines. In particular, we compare to various embedding based models such as DistMult [Yang et al., 2015], ComplEx [Trouillon et al., 2016] and ConvE [Dettmers et al., 2018]. We also wanted to compare to various neural models for learning logical rules such as neural-theorem-provers [Rocktäschel and Riedel, 2017], NeuralLP [Yang et al., 2017] and MINERVA [Das et al., 2018]. However, only MINERVA scaled to the size of NELL-995 dataset and others did not. As it is clear from table 2, CBR outperforms all the baselines by a large margin with gains of over 4% on the strict HITS@1 metric. We further discuss and analyze the results in sec (3.4).

**FB122**: Next we consider the FB122 dataset by Guo et al. [2016]. Comparing results on FB122 is attractive for a couple of reasons — (a) Firstly, this dataset comes with a set of logical rules hand coded by the authors that can be used for logical inference. It would be interesting to see if our CBR approach is able to automatically uncover the rules from the data. (b) Secondly, there is a recent work on a neural model for logical inference (GNTPs) [Minervini et al., 2020] that scales neural-theorem-provers [Rocktäschel and Riedel, 2017] to this dataset and hence we can directly compare with them. Table 1 reports the results. We compare with several baselines. CBR significantly outperforms GNTPs and even outperforms most models which have access to the hand-coded rules

| Metric | TransE | DistMult | ComplEx | ConvE | RotatE | GNTP | CBR |
|--------|--------|----------|---------|-------|--------|------|-----|
| HITS@1 | - | 0.39 | 0.41 | 0.40 | **0.43** | 0.41 | 0.39 |
| HITS@3 | - | 0.44 | 0.46 | 0.44 | **0.49** | 0.44 | 0.46 |
| HITS@10 | 0.50 | 0.49 | 0.51 | 0.52 | **0.57** | 0.48 | 0.51 |
| MRR | 0.23 | 0.43 | 0.44 | 0.43 | **0.48** | 0.43 | 0.43 |

Table 3: Link prediction results on WN18RR dataset.

| Model | HITS@1 | HITS@10 | MRR |
|-------|--------|---------|-----|
| NeuralLP [Yang et al., 2017] | 0.048 | 0.351 | 0.179 |
| NTP-λ [Rocktäschel and Riedel, 2017] | 0.102 | 0.334 | 0.155 |
| MINERVA [Das et al., 2018] | 0.162 | 0.283 | 0.201 |
| MultiHop(DistMult) [Lin et al., 2018] | 0.145 | 0.306 | 0.200 |
| MultiHop(ConvE) [Lin et al., 2018] | 0.178 | 0.329 | 0.231 |
| Meta-KGR(DistMult) [Lv et al., 2019] | 0.197 | 0.345 | 0.248 |
| Meta-KGR(ConvE) [Lv et al., 2019] | 0.197 | 0.347 | 0.253 |
| CBR (ours) | **0.234** | **0.403** | **0.293** |

Table 4: Link prediction results on NELL-995 for few shot relations.

during training. We also find that CBR is able to uncover correct rules for 27 out of 31 (87%) query relations.

**WN18RR**: Finally, we report results of WN18RR in table 3. CBR performs competitively with GNTPs and most embedding based methods except RotatE [Sun et al., 2019]. Upon further analysis, we find that for 210 triples in the test set, the entity was not present in the graph and hence no answers were returned for those query entities.

### 3.3 Experiments with limited data

As mentioned before, our CBR based approach needs no training and gathers reasoning patterns from few similar entities. Therefore, it should ideally perform well for query relations for which we do not have a lot of data. Recently, Lv et al. [2019] studied this problem and propose a meta-learning [Finn et al., 2017] based solution (Meta-KGR) for few-shot relations. We compare with their model to see if CBR based approach will be able to generalize for such few-shot relations. Table 4 reports the results and quite encouragingly we find that we outperform all sophisticated meta-learning approaches by a large margin.

### 3.4 Analysis: CBR is capable of doing contextualized reasoning

In this section, we analyze the performance of our non-parametric approach and try to understand why it works better than existing parametric rule learning models like MINERVA. Lets consider the relation 'agent_belongs_to_organization' in the NELL-995 dataset. Here the query entity can belong to a wide varity of types. For example, these are all triples for the query relation in NELL-995— (George Bush, agent_belongs_to_organization, House of Republicans), (Vancouver Canucks, agent_belongs_to_organization, NHL), (Chevrolet, agent_belongs_to_organization, General Motors). As can be seen, the query entity for a relation can belong to many different types and hence the logical rules that needs to be learned would be different. An advantage of CBR based approach is

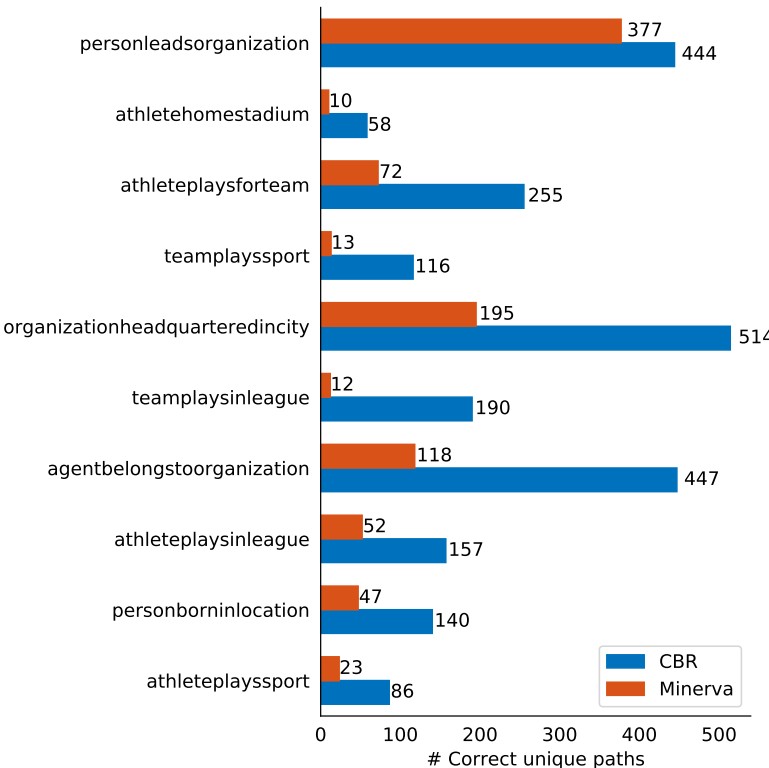

Figure 2: The number of unique correct paths CBR and MINERVA find for each query relation in NELL-995

that, for each query entity it retrieves similar contexts and then gather rules from them. One the other hand, models like MINERVA has to encode all rules into its parameters which makes learning harder. In other words, CBR is capable of doing better fine-grained contextual reasoning for a given query. To further confirm this hypothesis, we count the number of paths that CBR finds that lead to the correct answer and compare it with MINERVA. CBR learns a total of 306.4 unique paths that lead to answer compared to 176.83 of MINERVA. Figure 2 plots the counts for each query relation which further shows that CBR finds more varied paths than MINERVA.

### 3.5 Results with RotatE embeddings

As mentioned before, the CBR approach is generic and one can incorporate sophisticated models into each of the step. We run experiments where the *m*-hot representation of entities is replaced with pretrained embeddings obtained from a trained RotatE [Sun et al., 2019] model for building the similarity matrix. On the WN18RR dataset, we get a MRR of 0.425 as compared to 0.423 with our original approach.

### 3.6 Limitations of our Current Work

A key limitation of our current work is the symbolic matching of reasoning paths. Different symbolic sequence of relation can have similar semantics (e.g. works_in_org, org_located_in and studies_in,

college_located_in are similar paths for inferring lives_in relation), but in our current work, these paths would be treated differently. This limitation can be alleviated by learning a similarity kernel for paths using distributed representation of relations.

On error analysis, we found that a major source of error occurs in our ranking of entities. Currently, ranking of predicted entities is done via number of paths that lead to it. Even though, this simple technique works well, we noticed that, if we had access to an oracle ranker, our performance would improve significantly. For example, in FB122 dataset, CBR retrieves a total of 241 entities out of more than 9.5K entities. An oracle ranking of these entities would increase the accuracy (HITS@1) from 0.28 to 0.74. This indicates that a substantial gain could be obtained if we train a model to rank the retrieved entities, a direction we leave as future work.

### 3.7 Inference time

Table 5 report the inference times of our model on the entire evaluation set of WN18RR (6268 queries) and NELL-995 (2825 queries) and compares it with MINERVA. Since our approach first retrieves similar entities and then gathers reasoning paths from them, it is slower than MINERVA. However, given the empirical improvements in accuracy, we believe this is not a significant tradeoff.

| Dataset | MINERVA | CBR |
|---------|---------|-----|
| WN18RR | 63s | 69s |
| NELL-995 | 35s | 68s |

Table 5: Inference time (in seconds) on two datasets.

## 4. Related Work

**Bayesian non-parametric approaches for link prediction**: There is a rich body of work which employs bayesian non-parametric approaches to automatically learn the latent dimension of entities. The infinite relational model (IRM) [Kemp et al., 2006] and its extensions [Xu et al., 2006] learns a possibly unbounded number of latent clusters from the graph and an entity is represented by its cluster membership. Later Airoldi et al. [2008] proposed the mixed membership stochastic block models that allows entities to have mixed membership over the clusters. Instead of cluster membership, Miller et al. [2009] propose a model that learn features of entities and the non-parametric approach allows learning unbounded number of dimensions. Sutskever et al. [2009] combine bayesian non-parametric and tensor factorization approaches and Zhu et al. [2012] allows non-parametric learning in a max-margin framework. Our method does not learn latent dimension and features of entities using a bayesian approach. Instead we propose a framework for doing non-parametric reasoning by learning patterns from k-nearest neighbors of the query entity. Also, the bayesian models have only been applied to very small knowledge graphs (containing few hundred entities and few relations).

**Rule induction in knowledge graphs** is a very rich field with lots of seminal works. Inductive Logic Programming (ILP) [Muggleton et al., 1992] learns general purpose predicate rules from examples and background knowledge. Early work in ILP such as FOIL [Quinlan, 1990], PROGOL [Muggleton, 1995] are either rule-based or require negative examples which is often hard to find in KBs (by design, KBs store true facts). Statistical relational learning methods [Getoor and Taskar, 2007, Kok and Domingos, 2007, Schoenmackers et al., 2010] along with probabilistic logic [Richardson and Domingos, 2006, Broecheler et al., 2010, Wang et al., 2013] combine machine learning and logic but these approaches operate on symbols rather than vectors and hence do not enjoy the generalization properties of embedding based approaches. Moreover, unlike our approach, these methods do not learn rules from entities similar to the query entity. Recent work in rule induction, as discussed before

[Yang et al., 2017, Rocktäschel and Riedel, 2017, Das et al., 2018, Minervini et al., 2020] try to encode rules in the parameters of the model. In contrast, we propose a non-parametric approach for doing so. Moreover our model outperforms them on several datasets.

**K-NN based approach in other NLP applications**: Nearest neighbor models have been applied to a number of NLP applications in the past such as parts-of-speech tagging [Daelemans et al., 1996] and morphological analysis [Bosch et al., 2007]. There has also been several recent work which leverages k-nearest neighbors for various NLP tasks, which is a step towards case based reasoning. Retrieve-and-edit based approaches are gaining popularity for various structured prediction tasks [Guu et al., 2018, Hashimoto et al., 2018]. Accurate sequence labeling by explicitly and only copying labels from retrieved neighbors have been achieved by Wiseman and Stratos [2019]. Another recent line of work use training examples at test time to improve language generation [Weston et al., 2018, Pandey et al., 2018, Cao et al., 2018, Peng et al., 2019]. Improvements in language model have been also observed by Khandelwal et al. [2020] by utilizing explicit examples from past training data obtained from nearest neighbor search in the encoded space. However, unlike us these work do not extract explicit reasoning patterns (or solutions in cases) from nearest neighbors.

## 5. Conclusion

We propose a very simple non-parametric approach for reasoning in KGs that is similar to case-based reasoning approaches in classical AI. Our proposed model requires no training and can be readily applied to any knowledge graphs. It achieves new state-of-the-art performance in NELL-995 and FB122 datasets. Also we show that, our approach is robust in low-data settings. Overall, our non-parametric approach is capable of deriving crisp logical rules for each query by extracting reasoning patterns from other entities and entirely removes the burden of storing logical rules in the model parameters.

## Acknowledgements

This work is funded in part by the Center for Data Science and the Center for Intelligent Information Retrieval, and in part by the National Science Foundation under Grant No. IIS-1514053 and in part by the International Business Machines Corporation Cognitive Horizons Network agreement number W1668553 and in part by the Chan Zuckerberg Initiative under the project Scientific Knowledge Base Construction. Any opinions, findings and conclusions or recommendations expressed in this material are those of the authors and do not necessarily reflect those of the sponsor.

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
