# OpenReview forum: "A Simple Approach to Case-Based Reasoning in Knowledge Bases"
_AKBC.ws/2020/Conference — AKBC 2020_

### Official Review · AnonReviewer2 · 2020-03-27
**Good results, but lacking a discussion about the limitations**

**Rating:** 7
**Confidence:** 3

**Review:**

This paper proposes a non-parametric approach for reasoning on knowledge graphs.
The described approach CBR (Case-Based Reasoning) includes several steps including matching similar entities and extracting rules that are used during inference to select the right entity.
The approach is shown to be effective in knowledge base completion (FB122, WN18R) and query-answering (NELL-992), and yield better overall results than several competitive parametric approaches including TransE, DistMul, Complex, MINERVA, ASR, KALE, GNTPs.

Pros:
* Evaluation in terms of performance is sufficient - comparing to several approaches, on two tasks (KBC, Query answering) and three different knowledge-graphs.
* State of the art results over several recent, well-performing approaches.
* Analysis of “why the approach is better than one of the compared systems” - MINERVA is insightful
* The approach is simple, easy to understand, and its modules can easily be extended with recent ML/DL modules.
* The reasoning seems interpretable to some extent since there are actual rules that are retrieved and used.

Cons:
* The limitations of the approach are not discussed in detail:
** What is the inference time compared to other parametric approaches? Could you include these in the paper?
** With the current approach what are the limitations of the size of the graph in terms of the number of triples, entities, relations?

* The title of the paper seems too general. This is not the first paper to propose “non-parametric” approaches for reasoning over knowledge graphs, nor it is an overview paper. If the paper is accepted the title must be changed to a more specific one.

---

> ### Author Response · Authors · 2020-04-18
> **Re Review 3:**
>
> We thank the reviewer for the insightful comments.
> Limitations of the approach: We added a new section in the paper discussing limitations of our current approach (Sec 3.6). In a nutshell, they are (a) we currently do exact symbolic matching of reasoning paths. This definitely needs to low-recall since there are paths with different sequence of relations but are semantically similar, (b) ranking of retrieved entities --- currently, retrieved entities are ranked based on the number of paths that lead to the entity. Upon error analysis, we found that we can significantly improve performance if we learn to rank the retrieved entities.
>
> Inference times: We have added the inference times of our method in sec x,x of the paper. We are slightly slower than MINERVA -- a parametric rule-learning approach that learns to find reasoning paths in KG . For example on the NELL-995 dataset, our approach takes 51s to process 2.8K queries, whereas MINERVA takes 35s. However, as we have shown, our approach is much more accurate than MINERVA.
>
> Limitations on the size of the knowledge graph: We believe our method will scale to very large knowledge bases containing millions of entities. This is because our approach only deals with a few entities similar to the query entity (which could be the output of a fast KNN search). On the other hand, embedding based models have to learn parameters for all the millions of entities and it is unclear such methods will scale to such large graphs. Moreover, parametric rule-learning methods such as MINERVA will also likely fail to capture all rules in its parameters at that scale.
>
> Changing the paper title:
> Thanks for the suggestion. We are currently discussing this among ourselves. We are considering several options such as
> (1) A Simple Approach for Case Based Reasoning in Knowledge Bases.
> (2) Case Based Reasoning in Knowledge Bases.
> (3) An Efficient Non-Parametric Approach for Reasoning in Knowledge Bases.
> (4) Case Based Reasoning in Knowledge Bases --- A Simple, Efficient, Non-Parametric Approach towards Automated Reasoning.
> (5) Non-parametric Reasoning in Knowledge Bases

---

### Official Review · AnonReviewer3 · 2020-03-28
**Interesting Simple Approach, Strong Results**

**Rating:** 8
**Confidence:** 3

**Review:**

This paper provides a very simple approach to the problem of knowledge base completion. The idea is this - given a query (subject, relation), you find other entities similar to subject, see which other paths they can take to their corresponding object if they express the relation, and check if the subject express those paths. Object reached this way are candidate answers. The one with most paths reached is marked correct.

One question I have for author is that - when the new relation discovered is entered into memory, does it come with any form of weighting that tells us how confident the model is in its prediction. For example, if we take the (Melinda Gates, works in, ?), the expressed path (ceo, based in) may not be correct for this new subject. Perhaps, a discussion on this problem will make the paper stronger.

Why do we need caching ? Can't the paths be discovered in real time ? Can there be better heuristic designed that can be used to filter paths at test time (depending perhaps on the subject of the query itself) ?

The authors present results on multiple datasets where they are either SOTA or competitive (I cannot comment on this with complete confidence if the author has missed any other relevant comparisons). They also perform qualitative testing to see why their model has good performance. An error analysis on this model would also make this paper stronger.

In general, I like this approach for its simplicity (and generality as the authors note) and in hindsight, it seems surprising why this has never been tried before.

---

> ### Author Response · Authors · 2020-04-18
> **Re Review2:**
>
> We thank the reviewer for the insightful comments.
>
> Weights for model prediction: Because of the symbolic nature of our path matching model, our model does not give confidence scores and candidate answers are ranked by the number of paths that lead to them. Incorporating a model that is capable of aligning similar paths (even when they don’t match exactly) and aggregating them is one way of getting a confidence measure for model prediction. We leave this extension for future work.
>
> Need for caching: This is a great point. The main reason behind caching paths is for efficiency and as you point out correctly, the paths can indeed be discovered in realtime. There needs to be a cache entry for every edge in the knowledge graph which can become prohibitively large to store. For the same reason, in our current implementation, we exhaustively search for paths between two entity pairs in real-time in a small subgraph around an entity.
> We also agree with your suggestion that models such as DeepPath[1] or MINERVA[2] can be applied to automatically gather paths.
>
> Error analysis: As per your suggestion, we did error analysis and reported in Section 3.7 of the paper. Our main finding is that, a major source of error occurs in our ranking of entities. Currently, ranking of predicted entities is done via number of paths that lead to it. Even though this simple technique works well, we noticed that, if we had access to an oracle ranker, our performance would improve significantly. For example, in FB122 dataset, CBR retrieves a total of 241 entities out of more than 9.5K entities. An oracle ranking of these entities would increase the accuracy from 0.28 to 0.74. This indicates that a substantial gain could be obtained if we train a model to rank the retrieved entities. We leave this direction for future work.

---

### Official Review · AnonReviewer1 · 2020-03-28
**A simple but well-motivated method with strong performance**

**Rating:** 8
**Confidence:** 4

**Review:**

This paper proposes a non-parametric reasoning method for reasoning on incomplete knowledge bases. Specifically, for the task of finding a target entity given a source entity and a relation, since this specific relation might be missing for the source entity, multi-hop reasoning is required to get the answer. To get the reasoning paths, this paper proposes to first retrieve similar entities from the knowledge base that have the same outgoing relation, and then gather all possible reasoning paths from these retrieved entities. Finally, these reasoning paths extracted from other entities can be applied to the source entity in the query and get the answer.

The methods proposed in this paper is simple yet very effective. They outperformed previous strong models on NELL-995 and FB-122. Moreover, because of the non-parametric property, this method is also robust in low data settings. I also like the general thinking that instead of encoding all the reasoning rules into model parameters, the case-based reasoning system might worth more attention. I think this paper gave a good initial attempt and established a good framework for future work. For example, as the author mentioned in the paper, neural relation extraction systems can be incorporated to replace the exact-string matching. Therefore, I think this paper should be accepted.

---

> ### Author Response · Authors · 2020-04-18
> **Re Review1:**
>
> We thank the reviewer for insightful and positive comments. We agree with you that our work would serve as a framework for several future work in this direction. Thank you.

---

### Author Response · Authors · 2020-04-18
**Summary of changes**

We thank all reviewers for their insightful comments. We have made the following changes in the new version of our paper.

1. Change in scores of NELL-995 dataset: After the initial submission, we found a bug in our NELL-995 experimental setup. Specifically, there were some issues with test-time leakage. We fixed it and ran rigorous experiments. We have updated the results in Table 2. We are still state-of-the-art in all metrics compared to all our baselines, so the primary conclusion of the paper does not change at all.
2. We have added new analysis with inference times (Sec 3.6)
3. Based on the review comments, we have added a discussion on the limitation of the current approach (Section 3.7)

---

### Decision · Program_Chairs · 2020-05-01

**Decision:**

Accept

**Comment:**

This paper proposed a case-based reasoning/non-parametric approach for a widely studied knowledge base completion task: given a subject and a relation, predict the object based on a given knowledge graph. The idea is novel and simple: given the subject, it retrieves similar entities in the whole KG and corresponding reasoning paths with respect to the query relation and uses multiple paths of evidence to derive an answer. The approach has been evaluated on multiple benchmarks and demonstrates excellent performance.

All the reviewers think this is a strong paper and would lay out a solid framework for future work in this direction.  We  recommend accepting this paper.

As per the suggestions by the reviewers, it is a good idea to consider adding “case-based reasoning” to the title to reflect the key idea of this approach. It would be also be desired to discuss how this approach compares to other existing approaches (inference time, scalability, etc) in addition to accuracy metrics.